# Neurodynamic Treatment Promotes Mechanical Pain Modulation in Sensory Neurons and Nerve Regeneration in Rats

**DOI:** 10.3390/biomedicines10061296

**Published:** 2022-05-31

**Authors:** Giacomo Carta, Benedetta Elena Fornasari, Federica Fregnan, Giulia Ronchi, Stefano De Zanet, Luisa Muratori, Giulia Nato, Marco Fogli, Giovanna Gambarotta, Stefano Geuna, Stefania Raimondo

**Affiliations:** 1Department of Clinical and Biological Sciences, University of Torino, 10043 Torino, Italy; giacomo.carta@unito.it (G.C.); benedettaelena.fornasari@unito.it (B.E.F.); giulia.ronchi@unito.it (G.R.); stefano.dezanet@edu.unito.it (S.D.Z.); luisa.muratori@unito.it (L.M.); giovanna.gambarotta@unito.it (G.G.); stefano.geuna@unito.it (S.G.); stefania.raimondo@unito.it (S.R.); 2Neuroscience Institute Cavalieri Ottolenghi (NICO), University of Torino, 10043 Torino, Italy; giulia.nato@unito.it (G.N.); marco.fogli@unito.it (M.F.); 3Department of Rehabilitation, ASST (Azienda Socio Sanitaria Territoriali) Nord Milano, Sesto San Giovanni Hospital, Sesto San Giovanni, 20099 Milano, Italy; 4Department of Life Sciences and Systems Biology, University of Torino, 10124 Torino, Italy

**Keywords:** neuropathic pain, neurodynamic, nociceptor, allodynia, regeneration, motor recovery, non-pharmacological treatment, dorsal root ganglia, gene expression

## Abstract

Background: Somatic nerve injuries are a rising problem leading to disability associated with neuropathic pain commonly reported as mechanical allodynia (MA) and hyperalgesia. These symptoms are strongly dependent on specific processes in the dorsal root ganglia (DRG). Neurodynamic treatment (NDT), consisting of selective uniaxial nerve repeated tension protocols, effectively reduces pain and disability in neuropathic pain patients even though the biological mechanisms remain poorly characterized. We aimed to define, both in vivo and ex vivo, how NDT could promote nerve regeneration and modulate some processes in the DRG linked to MA and hyperalgesia. Methods: We examined in Wistar rats, after unilateral median and ulnar nerve crush, the therapeutic effects of NDT and the possible protective effects of NDT administered for 10 days before the injury. We adopted an ex vivo model of DRG organotypic explant subjected to NDT to explore the selective effects on DRG cells. Results: Behavioural tests, morphological and morphometrical analyses, and gene and protein expression analyses were performed, and these tests revealed that NDT promotes nerve regeneration processes, speeds up sensory motor recovery, and modulates mechanical pain by affecting, in the DRG, the expression of TACAN, a mechanosensitive receptor shared between humans and rats responsible for MA and hyperalgesia. The ex vivo experiments have shown that NDT increases neurite regrowth and confirmed the modulation of TACAN. Conclusions: The results obtained in this study on the biological and molecular mechanisms induced by NDT will allow the exploration, in future clinical trials, of its efficacy in different conditions of neuropathic pain.

## 1. Introduction

Neuropathic pain and disability are common conditions in patients suffering from peripheral nervous system (PNS) injuries or pathologies [1,2]. In particular, when PNS injuries occur, they are often associated with sensory and motor impairment or loss of function distal to the lesion site. Unfortunately, the interventional procedure improvement and opioid prescriptions did not reduce the burden of neuropathic pain, which is still a clinical challenge in terms of making a diagnosis and defining and maintaining an adequate therapy [3,4,5].

After a nerve injury, unmyelinated and myelinated primary afferent neurons increase their activity, sustaining a predominantly peripheral neuropathic pain process with stimuli that are dependent on abnormal responses such as mechanical allodynia (MA) and hyperalgesia [6,7,8]. These conditions are related respectively to a pain perception for non-painful pressure or stretch and exaggerated pain for painful stimuli. These processes are linked to changes in neurochemical, anatomical, electrophysiological properties or protein and gene expression in the dorsal root ganglia (DRG) neurons [8,9,10,11,12,13]. More in detail, MA and hyperalgesia are related to the expression of specific genes in the DRG, such as the mechanosensitive ion channel *TACAN* (Tmem120A) [10,12,14], or the Toll-like receptor-2 (*TLR2*), a receptor modulating the macrophage response [13,15,16,17]. *TACAN* is a mechanical high threshold ion channel that can be activated only by a cell surface tension higher than 35 mN/m and is responsible for mechanical painful stimuli transduction [18], differently to the PIEZO mechanosensitive ion channels which are activated by 1.4–5 mN/m forces and are responsible for touch and non-painful pressure sensations [19]. *TLR2* has been selectively linked by Cobos and colleagues to MA and not cold allodynia in the DRG neurons in a neuropathic pain model due to nerve mechanical injury [15]. Notably, these mechanisms causing MA and hyperalgesia related to neuropathic pain are highly conserved in both murine and human DRG neurons [12,20,21,22]. No specific drug to modulate *TLR2* or *TACAN* expression in the DRG is now available on the market, but the modulation of receptors and mechanosensitive channels expressed by peripheral sensory neurons has been demonstrated to be a promising strategy in the treatment of neuropathic pain [10,16,23,24]. The depletion or suppression of *TACAN* and *TLR2* suppresses MA and hyperalgesia [10,12,15,16]. In addition, RNA-Seq studies on DRG after nerve injuries in rats revealed that these genes are expressed regardless of sex [25,26,27].

The selective repeated administration of uniaxial tensions to somatic nerves, also known as neurodynamic treatment (NDT), has shown to be effective in pain modulation and nerve conduction improvements detected by electrophysiological assessment in long term neuropathic pain patients suffering from carpal tunnel syndrome [28,29,30,31] and radiculopathies of somatic nerves in the upper or lower limbs [32,33,34,35,36,37]. NDT significantly reduces pain and disability with long-term effects in neuropathic pain patients. In addition, hypoalgesic and anti-allodynic short-term effects have been reported respectively in asymptomatic [37,38,39,40] and neuropathic pain patients [41,42,43]. Despite the effectiveness of NDT in modulating neuropathic pain, the biological mechanisms induced by this treatment remain poorly characterized. Our previous studies have shown that NDT induces dose-dependent pro-regenerative and antiapoptotic effects on sensory and motor neuronal cell lines [44], but as far as we know, no data on MA and hyperalgesia modulation are available. In the current study, we aimed to investigate in vivo if the NDT modulates neuropathic pain and promotes motor sensory recovery after a nerve crush injury [45,46]. To further explore the relevance of possible pain modulation processes induced by NDT in the DRG, we adopted an ex vivo model of DRG organotypic explant which was subjected to neurodynamic treatment through a bioreactor previously validated for this purpose [44].

## 2. Materials and Methods

### 2.1. Animals

#### 2.1.1. In Vivo Surgical and Treatment Protocols

Adult female Wistar rats approximately weighing 230–250 g (Charles River Laboratories, Milan, Italy) were used in the present study, and a proper sample size (n = 18; 6 for each experimental group) was calculated based on grasping strength (370 ± 90 g; η_p_^2^ = 0.14 large) after a median nerve crush injury in rat upper limbs reported by Ronchi and colleagues [47] (effect size f = 0.403; power (1 − β err prob) = 0.95; α = 0.05; actual power = 0.97). Only female animals were adopted since they maintain, in adulthood, a limited weight and size, and this allows for an easier handling during NDT administration and functional testing. Furthermore, RNA-Seq studies on DRG after nerve injuries in rats have shown that the genes related to mechanical allodynia that we have assessed in our study are similarly expressed both in males and females [25,26,27]. Animals were housed in large cages at the animal facility of Neuroscience Institute Cavalieri Ottolenghi (NICO; Ministerial authorization DM 182 2010-A 3–11-2010) in humidity and temperature-controlled room with 12 h light/dark cycles. Free access to water and standard chow was provided Humane endpoint criteria were adopted to adequately measure and minimize any animal pain, discomfort, or distress. The study conditions conformed to the guidelines of the European Union’s Directive EU/2010/63. In addition, approval by the Ethic Experimental Committee of the University of Turin (Ministry of Health project number 864/2016) was obtained before the research began. As reported in Figure 1A, animals were randomly divided into 3 experimental groups: (i) 6 rats subjected to crush injury but not treated, used as control (NO NDT); (ii) 6 rats subjected to crush injury and treated with an NDT protocol from the third day after surgery until day 24 (NDT POST; described further below); (iii) 6 rats subjected to crush injury and treated with NDT both 12 days before nerve injury (5 days/week) and for 24 days after injury with the same posology of the NDT POST group (NDT PRE + POST).

The surgical interventions were carried out under deep anaesthesia using Tiletamine (052019, Bayer, Milano, Italy) + Zolazepam (Zoletil) (A101580025, Virbac, Milano, Italy) (40 and 5 mg/kg). After trichotomy, the median and ulnar nerves of the left forelimb were exposed from the axillary region of the arm above the elbow (Figure 2A). The nerve crush injury was performed by administering a pressure of 17.02 MPa for 30 consecutive seconds [47] to the middle third of the nerves using a flat tip clamp as reported in Figure 2B,C.

The protocol of NDT tensioning was administered once a day, with 1 s of stretch (till resistance was felt by the clinician), and 5 s to return to the starting position 30 times [44,48], within the time window from 1:00 pm to 3:00 pm 5 days/week until day 24 post-injury (Figure 2D–H). The rat neck was kept in contralateral lateral side flexion (right side flexion) by gently and firmly surrounding its neck with the right thumb and index fingers. The clinician gently brought the shoulder into abduction, elbow in extension, and wrists and fingers in extension by gently pinching the paw with the left thumb on the back of the paw and abducting the left index finger. The progression of the manoeuvres was performed until resistance was perceived. The perception of passive tissue resistance is a standardized and reliable hallmark for NDT administration reported in clinical and preclinical studies [44,49,50,51]. The rats were awake during treatments and were free to withdraw their paw anytime during the manipulations and experimental tests.

#### 2.1.2. Behavioural Tests

All reported behavioural tests were administered four days before the nerve crush and 4, 8, 12, 15, 19, and 22 days after the nerve injury with the same chronological order as they are listed below, starting from sensory tests. The scores obtained before nerve injury, from sensory and motor behaviour tests, were adopted as a reference to define the physiological normal response.

##### Sensory Assessment


Von Frey Test


The Von Frey [52] calibrated monofilaments were adopted to define non-noxious, noxious pressure threshold before nerve injury, and anaesthesia or MA after the injury. The touch threshold (non-noxious pressure threshold) was defined by observing the lack of paw movement or withdrawal when the filament was bending and the rat was keeping its weight on the assessed paw. The noxious pressure or mechanical pain threshold was defined as the presence of paw withdrawal also associated with licking and flicking behaviours. The anaesthesia for pressure or puncture painful stimuli was defined as the lack of reactions related to stimuli administration after the nerve injury. MA was defined by observing paw withdrawals elicited by the pressure that before the nerve injury induced no paw withdrawals. Animals were acclimatized individually for 15 min in a Plexiglas box (28 × 20 × 14 cm^3^) with a metal grid on the bottom (0.3 × 0.4 cm^2^). The assessment was started for each side with a filament with 5 g (g) bending force applied with uniform pressure to the palmar surface of the paw for 5 s. A progressive series of 5, 10, 15, 20, and 25 g was applied until a change in response pattern was elicited and confirmed by 4 additional applications of the previous pressure eliciting no responses and 4 of the reached pressure eliciting responses [53]. When rats did not respond to any filament, a score of 45 g was assigned [54].


Pinprick Test


A modified pinprick test score on pain [55], assessing the withdrawal responses to 4 consecutive pin administration to the hind paw was adopted as follows: “0” for the absence of withdrawal response, “1” when the rat slowly removes the paw from the pin, “2” when the paw was quickly removed, and “3” when the rat quickly removed the paw flicking and licking it.

##### Nerve Irritability to Tension

Twenty-two days after the injury, mechanical nerve irritability was assessed at the injured side by recording the number of paw withdrawals during 30 consecutive nerve tensions administered until resistance was perceived by the clinician. Tensions were administered at 0.5 Hz frequency (five times higher rate than treatment protocols; 0.5/s tension/release) able to induce a windup or temporal summation phenomenon, consisting of transiently increased responsiveness of spinal neurons to nociceptive peripheral fibres repeated stimulations [56,57]. This test assessed the performance of the pain descending inhibitory system and central sensitization to pain, processes that NDT possibly affects as reported in the literature [58,59].

##### Motor Performances


Grissini Test


The forepaw dexterity was assessed and quantitatively measured through the modified Cappellini handling test described by Tennent and colleagues [60], using “grissini”, typical Italian breadsticks. In brief, a 70 mm long grissini piece (Ø 6 mm diameter; Roberto S.r.L. Treviso, Italy), was given to the rat, and the time to eat the entire piece and the number of adjustments (defined as any observed replacement or removal of the paw on the grissini after it started eating) were videorecorded and later analysed by observing slowed videos at ~50% of real-time speed.


Rope Test


The global strength of upper limbs and coordination were quantitatively assessed through the rope climbing test [61,62]. The rat was positioned at the bottom of the vertical 160 cm rope and was persuaded to climb it by gently touching the tail. The time from the climbing of the rat starts until the crossing of the entrance on the top platform (40 × 40 cm^2^) was recorded. Rats were gradually trained to climb for 10 days before the surgery, as described by Diogo and co-workers [55], with a daily increment of 20 cm rope.


Grasping Test


The grasping test was performed to assess the finger flexor strength as described by Papalia and colleagues [63] by holding the rat by its tail and lowering it towards the grid of the BS-GRIP Grip Meter device (GT3, 2Biological Instruments, Varese, Italy) until a firm grip was observed, and then the rat was pulled upwards until the grip was lost and the highest force was recorded by the device. The mathematical mean value obtained by three consecutive attempts was used for statistical analysis.

#### 2.1.3. Morphological and Morphometrical Analysis of Nerves

After 24 days from the nerve crush, animals were sacrificed by an intraperitoneal Zoletil (Virbac, A101580025, Milano, Italy) + Xilazina (Bayer, 052019, Milano, Italy) and excised 5 mm above and 10 mm below the middle third of their brachial portion (site of injury for the left side), and DRGs of the brachial plexus of both sides were excised, and samples were harvested as described below. Then, the *flexor digitorum superficialis* and *profundus* tendons were cut, and the entire muscle bellies were excised from the surrounding soft tissues and muscles. Finally, the muscle wet mass was weighted using an electronic analytical balance with a precision of 0.1 g [48].

##### Resin Embedding and High-Resolution Light Microscopy

As previously described [47], a segment of the median and ulnar nerves distally to the injury site was removed for the high resolution light microscopy. The same nerve portion (middle third of the brachial part) was also removed from the not injured side. As previously described [64], the proximal stumps were marked with a 7/0 stitch (Péters Surgical, 26S03B, Bobigny, France), and the nerves were fixed in 2.5% purified glutaraldehyde and 0.5% saccharose in 0.1 M Sorensen phosphate buffer for 5–6 h at 4 °C (Electron Microscopy Science, 16210, Hatfield, PA, USA). The samples were washed in a 1.5% saccharose in 0.1 M Sorensen phosphate buffer solution and post-fixed in 2% osmium tetroxide (Electron Microscopy Science, 19170) for 2 h. The dehydration of the samples was performed with cycles of 5 min in ethanol from 30° to 100° and two cycles of 7 min in propylene oxide (Sigma Aldrich, 32221, 11025, Milano, Italy).

After 1 h in a 1:1 mixture of propylene oxide and Glauerts’ mixture of resins overnight, samples were embedded in Glauerts’ mixture of resins (made of equal parts of Araldite M and Araldite Harter, HY 964, Sigma Aldrich, 10951). In the resin mixture, 0.5% of the plasticizer dibutyl phthalate (Sigma Aldrich) was added. For the final step, 2% of accelerator 964 was added to the resin to promote the polymerization of the embedding mixture, at 60 °C.

A 2.5 μm thick semi-thin transverse section was cut (Leica Microsystems Ultracut UCT ultramicrotome, Wetzlar, Germany) at the distal stump of each sample and stained using Toluidine blue for high-resolution light microscopy and stereological analysis. The following parameters were assessed: (1) total number of myelinated fibres, (2) myelinated fibre density; (3) diameter of myelinated fibres and axons; (4) myelin thickness; (5) axon diameter/fibre diameter *ratio* (g-*ratio*); and (6) intraneural collagen *ratio*, which was obtained by subtracting the total cross-sectional nerve area from the total area of nerve fibres assessed. The nerve section images used for the analysis were randomly selected through a 2D dissector probe to obtain unbiased representative samples of myelinated nerve fibres and analysed by an assessor blinded to the treatment and side of the sample following the method previously described [65].

An estimated sample of 33 random pictures from each protocol, 6 from each sample (n = 99, effect size f = 0.5; power (1 − β err prob) = 0.95; α = 0.05; actual power = 0.95) was evaluated based on the effect of crush injury on nerve morphology previously reported [47].

#### 2.1.4. Dorsal Root Ganglia Analysis

To confirm that the NDT effects on rat sensory performances were induced by processes modulated by the peripheral sensory neurons, the NDT protocol was performed on DRG explants.

Rats (n = 6, 250–280 g) were sacrificed by a lethal anaesthetic overdose of Zoletil (A101580025, Virbac, Milano, Italy) + Xilazina (052019, Bayer, Milano, Italy) (>60 and >10 mg/kg) by intraperitoneal injection. The vertebral spine from C3 to L5 was surgically dissected and the vertebral body and spinal processes were removed to gain ventral access to the spinal cord. Under the operative microscope, DRG were localized in the intraforaminal region, removed, isolated, and collected in a 100 × 20 mm^2^ Petri dish with 6 mL of F12 medium (Gibco, Carlsbad, CA, USA, 21127022). The fine forceps and fine blades were used to remove the pre- and post-ganglionic branches and to reduce the fibrous capsule surrounding the ganglia to allow the neurites extension. The DRG were seeded on type I collagen pre-coated silicon membranes (Flexcell^®^, SFM-C) [66] and incubated at 37 °C for 150 min in a 40 µL drop of Geltrex Matrigel (Thermo Fisher Scientific, Waltham, MA, USA, A1413302) diluted 1:1 in serum-free medium (SFM) [67] containing 50 ng/mL nerve growth factor (NGF; Invitrogen, Karlsruhe, Germany, A42578) and 1 ng/mL Vitamin C (Sigma-Aldrich, Saint Louis, MO, USA, A5960). Finally, 3 mL of differentiation medium (SFM with 50 ng/mL NGF; 1 ng/mL Vitamin C) was added to cover Matrigel/SFM drops on the silicon membranes in a 60 × 15 mm Petri dish (BD Falcon, 353004, NY, USA); DRG were maintained at 37 °C with 5% CO_2_.

### 2.2. Ex Vivo NDT Protocols

The DRG seeded on the silicone membranes were incubated for 48 h in differentiation medium, then moved into a previously validated manual bioreactor [44], and treated with the protocol of repeated uniaxial tensions (NDT). After the NDT protocol was administered, the membranes were returned to the differentiation medium until the experiment ended (48 h after the treatment administrations and 96 h from seeding the DRG on the membranes).

As described in our previous study [44], an NDT protocol with 1 s of stretch (strain rate of about 0.1 and 1% of the membrane rest length, 0.01–0.36 mm), and 5 s to return at the starting position for 30 times was adopted. This protocol was the same daily administered for the in vivo experiments of this study.

A sham control named “control out” (CTR OUT) was designed to assess the environmental effects of nourishment privation induced by the treatment protocols reported above. It consisted of keeping the membranes with DRG off from the medium for 180 s, corresponding to the time in which NDT treated membranes with DRG were positioned in the bioreactor [68]. Finally, a “control in” (CTR IN) protocol was determined, leaving the membranes in the medium for the duration of the entire experiment. The changes induced by the NDT linked to pain suppression and nerve regeneration, such as modulation of genes linked to MA, neuropathic pain, neurite outgrowth, and cell survival, were defined as “positive effects”. On the contrary, the changes linked to pain promotion and cell apoptosis were defined as “negative effects”. An estimated sample of 10 membranes for each protocol (effect size f = 0.835; power (1 − β err prob) = 0.95; α = 0.05; actual power = 0.96) was evaluated based on the NDT effects on neurite length of nociceptive sensory neurons [44].

#### Immunohistochemistry

The DRG explants were fixed in 4% PFA for 15 min, and a previously described protocol was adopted [44], incubating overnight with primary antibody anti-βIII-tubulin (mouse, monoclonal, 1:1000, SIGMA-ALDRICH, T8660) diluted in PBS and secondary antibody goat anti-mouse Alexa Fluor488 (1:200, Molecular Probes, A11029, Eugene, OR, USA) diluted in PBS. Explants were mounted with a Dako fluorescence mounting medium, the silicon membrane long edges were mounted parallel to the coverslip long edges [44].

The DRG explant neurites were evaluated by adopting the Sholl analysis method [69].

Images were acquired using a Leica SP5 confocal microscope (Leica Microsystems) equipped with a 40× oil immersion objective (HCX PL APO lambda blue 40.0 × 1.25 OIL UV). From the ImageJ plugin *Neurite-J* described by Torres-Espín and colleagues, the parameters considered were: the distance of the longest neurite (Dmax), the maximum number of neurites (Nmax), and the Sholl critical value, defined as the distance from the organotypic culture centre where the maximum number of interception was detected [69].

### 2.3. Quantitative Real-Time PCR (qRT-PCR) Analysis

For in vivo experiments, a pool of DRG from the brachial plexus of the crushed and of the not injured side (from each animal) was collected immediately after animal sacrifice and immediately frozen in dry ice and stored at −80 °C.

For ex vivo experiments, the medium was removed from the petri dish, membranes were immediately scraped with TRIzol Reagent (ThermoFisher, 15596026, Monza, Italy) and DRG were collected in DNA LoBind tube 1.5 mL (Eppendorf, EP0030108051, Milano, Italy). Then, 500 μL TRIzol was used for each sample, and samples were immediately frozen in dry ice and stored at −80 °C.

Total RNA was extracted with TRIzol Reagent according to the manufacturer’s instructions, and 0.75 μg RNA/sample was retro-transcribed. For in vivo experiments, the calibrator for the relative quantification was the average of the samples obtained from the not injured side in the NO NDT group, while for ex vivo experiments, the calibrator was the average of the CTR IN samples. TATA-box binding protein (TBP) was adopted as a reference gene [44]. PCR primers are reported in Table 1.

### 2.4. Western Blot

After RNA extraction with TRIzol Reagent, proteins were extracted following the manufacturer’s instructions [70], and protein pellets were dissolved in Laemli buffer (2.5% sodium dodecyl sulphate, 0.125 M Tris-HCl, pH 6.8) at 100 °C. The Bicinchoninic Acid assay kit (Sigma-Aldrich, B9643, Milano, Italy) was used to determine protein concentration. Then, 50 μg protein for each sample was resolved onto a 12% polyacrylamide denaturing gel; Western blot analysis was performed as previously described [44]. Primary and secondary antibodies are listed in Table 2. Bands were quantified through Image Lab version 6.1.0 build 7 software (2020; Bio-Rad Laboratories, Inc.). For in vivo experiments, data were expressed relative to the mean value of the quantified bands belonging to the healthy nerve (obtained from the not injured side of the “NO NDT” samples), while for ex vivo experiments, data were expressed relative to the mean values of the quantified bands of “CTR IN” samples.

### 2.5. Statistical Analysis

R Statistical Software (Foundation for Statistical Computing, Vienna, Austria) was adopted to perform statistical analyses [71]. The analyses were performed using the *rstatix* [72] and *car* [73] packages, and the *ggpbur* [74] package was used to perform plots reported in the figures.

The data normality was assessed through the Shapiroؘ–Wilk and Levene test, and if normality assumption was detected, data from in vivo experiments were assessed using a Two-way analysis of variance (ANOVA) considering the factors treatment X side. For data obtained from the behavioural analysis, the factors time X treatment were considered. One-way ANOVA was used to assess differences between continuous variables from ex vivo experiments, and Tukey post hoc correction was used for between-group comparisons. The Dunn Kruskal–Wallis multiple comparisons with Bonferroni post hoc method was adopted for not normally distributed variables to assess between-groups comparisons. The partial eta squared (η_p_^2^) was adopted to estimate the effect size for normally distributed variables (η_p_^2^; 0.010 = small effect; 0.059 = medium, and 0.138 = large effect) otherwise epsilon squared (ε_p_^2^) was used for non-normally distributed variables (effects: small <0.08, medium 0.08–0.26, large >0.26) [75,76,77].

Confidence intervals (CI) at 95% were calculated, and a statistical significance level of 0.05 was adopted. Values were expressed as mean ± SD (standard deviation). The level of significance was set at *p* < 0.05 (*), *p* < 0.01 (**), *p* < 0.001 (***), and *p* < 0.000 (****).

## 3. Results

### 3.1. Changes in Animal Behaviour Tests Induced by the Neurodynamic Treatment

To assess the possible positive and protective effects of NDT on peripheral nerve after injury, promoting nerve function recovery and pain modulation, different NDT protocols were followed, as described in materials and methods (Figure 1).

Four days before nerve injury, sensory and motor behaviour tests were performed, and the baseline performance homogeneity among all groups was assessed to detect possible differences (see Appendix A). No sensory test scores significantly differed from each experimental group. Among motor tests, only the time spent on eating the 7 cm breadstick (grissini test speed) reported a significant difference between the NDT PRE-POST and the NO NDT group (Appendix A; F (2,15) = 4.25; *p* = 0.035).

The behavioural tests were then performed 4, 8, 12, 15, 19, and 22 days after surgery. Results are reported in Figure 3 and Figure 4. Results of the two-way ANOVA statistical analysis are reported in Appendix A.

### 3.2. Sensory Changes

Two-way ANOVA revealed that the overall mechanical pain threshold (Figure 3A) at the injured side was significantly lower in the NDT POST and NDT PRE + POST groups compared to the NO NDT group (*p* = 0.000; 95% CI: 3.59–10.45; *p* = 0.003; 95% CI: 1.33–8.19). In particular, the NDT POST group showed a significant treatment effect on days 8 and 12 with a lower mechanical pain threshold compared to the NO NDT group, which had shown a lack of mechanical pain responses until day 15 after surgery (*p* = 0.000; 95% CI: 6.85–34.81; *p* = 0.000; 95% CI: 18.52–46.48). In addition, the NDT PRE + POST compared to the NO NDT group showed a significant interaction between the 2 factors time and treatment effect at 12 days (*p* = 0.000; 95% CI: 1.33–8.19). The NO NDT group reported a significantly lower mechanical pain threshold at 19 days compared to the NDT PRE + POST group (*p* = 0.048; 95% CI: 0.43–14.6), which can be described as mechanical allodynia. No significant changes were detected in the mechanical pain threshold on the not injured side (Appendix A).

The overall touch threshold at the injured side (Figure 3B) was significantly lower in the NO NDT group compared to the NDT POST and NDT PRE + POST groups at 8 and 15 days post-injury (*p* = 0.024; 95% CI: 0.22–3.83; *p* = 0.0005; 95% CI: 1.17–4.78). The touch threshold was also measured at the contralateral not injured side and a significant overall lower threshold (*p* = 0.004; 95% CI: 0.56–3.49) in the NO NDT compared to the NDT PRE + POST group was detected (data not shown).

The Pain Behaviour Scale (Figure 3C) showed that normal pain behaviours after the nerve injury were restored significantly early in the NDT POST and NDT PRE + POST groups compared to the NO NDT group. The NO NDT group lacked a normal pain response until 15 days after the nerve crush. Furthermore, the NDT POST group had a significant effect at 8 and 12 days with higher scores (closer to the normal pre-surgery values) than the NO NDT group (*p* = 0.0003; 95% CI: 0.23–1.43; *p* = 0.000; 95% CI: 0.23–1.43). In addition, a significant effect was detected for the NDT PRE + POST at 12 days with a higher score than the NO NDT group (*p* = 0.0003; 95% CI: 0.23–1.43).

Finally, the number of withdrawals for repeated high frequency neurodynamic tests to assess descending pain modulation effects (Figure 3D) measured 22 days after injury was significantly lower in the NDT POST and NDT PRE + POST groups compared to the NO NDT group (F (2,15)= 6.894, *p* = 0.008; 95% CI: 0.43–5.57 and 0.76–5.90; η^2^_p_ = 0.479, very large effect).

### 3.3. Motor Changes

The speed of eating the 7 cm grissini revealed no significant interaction between time and treatment factors, as shown in Figure 4A. An overall significantly lower speed was detected comparing the NDT PRE + POST to the NO NDT group (Appendix A; *p* = 0.007; 95% CI: 3.34–24.83). The number of paw adjustments performed while eating the grissini (Figure 4B) was not significantly different among groups and within time (Appendix A).

A significant overall effect for treatment was detected on the climbing speed (Figure 4C and Appendix A) between NDT POST and NO NDT group and NDT PRE + POST and NO NDT group (*p* = 0.009; 95% CI: 5.46–46.97; *p* = 0.0005; 95% CI: 13.26–54.77). Finally, no significant differences were detectable among groups at the grasping test (Figure 4D), where all animals showed signs of grip strength recovery starting from 12–15 days after the nerve injury. Finally, the rat total weight and the flexor digitorum superficialis and profundus wet muscle weight were not significantly different among all groups (Appendix A).

### 3.4. High-Resolution Light Microscopy

To study the possible effect of NDT on peripheral nerve regeneration, we performed stereological and morphometrical analysis on both the “healthy” median nerves (the contralateral nerve, not injured) and the regenerated nerves, 24 days after the crush lesion.

From a morphological point of view, the regenerated median nerves belonging to the three experimental groups showed the typical aspect of nerve regeneration with smaller nerve fibres and thinner myelin sheath compared to contralateral not injured nerves (Figure 5A). Quantitative analysis showed a significant higher total number of myelinated fibres in the injured side compared to the not injured side (Figure 5B) only for the NO NDT group (F (1, 156) = −17.67; *p* = 0.02; 95% CI: −2214.93–−371.85; η^2^_p_ = 0.102, medium effect), while no significant effect of side x treatment were detected in the two treated groups (F (1, 156) = 0.001; *p* = 0.999; η^2^_p_ = 0.000, very low effect). The axon and fibre diameter were significantly different between the two sides for all the experimental groups (Figure 5D,E), without any significant difference between the treatment groups (F (1, 156) = 195.95; *p* = 0.000; 95% CI: −2.36–−1.77; η^2^_p_ = 0.557, very large effect). The same results were observed in myelin thickness evaluation (Figure 5F), with thinner myelin on the injured side compared to the not injured side, for all experimental groups (F (1, 156) = 266.38; *p* = 0.000; 95% CI: 2.69– 3.43; η^2^_p_ = 0.631, very large effect). A significant difference between sides in the NO NDT group was reported for the g-*ratio* values (F (1, 156) = 16.68; *p* = 0.0001; 95% CI: 0.68– 0.71; η^2^_p_ = 0.097, small effect; Figure 5G), and a significant higher g-*ratio* between injured NO NDT and NDT POST groups was observed (F (2, 156) = 6.74; *p* = 0.0015; η^2^_p_ = 0.041, small effect). Finally, a significant difference was detected between the injured and not injured sides in all groups for intraneural fibrosis (F (1, 156) = 239.27; *p* = 0.0001; 95% CI: 41.62–52.75; η^2^_p_ = 0.605, very large effect; Figure 5H). Notably, the NDT POST group reported a significant lower ratio of intraneural collagen compared to the NO NDT group at the injured side (F (1, 156) = 1.99; *p* = 0.02; 95% CI: 0.01–0.21; η^2^_p_ = 0.025, small effect).

### 3.5. Gene and Protein Expression

Gene expression analysis on DRG of both ipsilateral (injured) and contralateral (not injured) sides was performed 24 days after nerve injury to detect any differences between sides and to assess possible systemic NDT effect. Genes linked to MA mediated by the immune response (*TLR2*) and by the mechanosensitive channels induced by TNFα (TACAN) were evaluated by comparing the treatments and sides from in vivo experiments (Figure 6A). No significant treatment effect was detected for the *TLR2* gene expression (F (2,30) = 1.00; *p* = 0.38; η_p_^2^ = 0.063, small effect; F (2,18) = 0.673; *p* = 0.418; η_p_^2^ = 0.022, small effect). A significant lower *TACAN* expression was detected between NDT POST and NO NDT groups, both in the injured and not injured sides (F (2,30) = 1.002; *p* = 0.02; 95% CI: 0.01 – 0.18; η_p_^2^ = 0.063, small effect; and F (2,30) = 5.08; *p* = 0.03; 95% CI: 0.01 – 0.28; η_p_^2^ = 0.022, small effect). Finally, in the NO NDT group, a significant difference was also observed between the injured and not injured sides (Figure 6A; F (1,30) = 1.002; *p* = 0.04; η_p_^2^ = 0.022, small effect), while such a difference was not observed in the other two experimental groups.

The protein expression of *TLR2* and *TACAN* was also evaluated in the DRG comparing the treatments and sides, as represented in Figure 6B. No significant differences between treatment and sides were detectable at protein level for *TLR2* and *TACAN* (F (2,18) = 0.087; *p* = 0.91; η_p_^2^ = 0.010, small effect; and F (2,18) = 0.012; *p* = 0.99; η_p_^2^ = 0.001, very small effect).

### 3.6. Neurodynamic Effects on Organotypic Cultures of DRG Explants

To further confirm that the NDT protocol induced selective DRG neuron responses, ex vivo cultures of DRG explants from healthy rats were subjected to NDT (30 repetitions) using a previously validated manual bioreactor [44], and analyses were performed two days after the treatment (Figure 7). A significant NDT effect was observed on the neurite length (Dmax; F (2,27) = 5.14; *p* = 0.005; η_p_^2^ =0.3, large effect). A significantly higher Dmax was detected between NDT and both the CTR OUT and CTR IN groups (Figure 7B and Appendix A). No significant differences were detected for the maximum number of neurites (Nmax) and Sholl critical value among the three experimental groups (Appendix A and Figure 7B).

Gene expression analysis showed no significant differences for the expression of *TLR2* (Figure 7C), (F (2,26) = 2.08; *p* = 0.145; 95% CI: 0.46–1.01; η_p_^2^ = 0.138, medium effect). A significant effect of treatment was detected for *TACAN* expression (F (2,26) = 5.28; *p* = 0.0132; 95% CI: 0.66–0.91; η_p_^2^ = 0.289, large effect). In particular, a significant downregulation of *TACAN* expression was detected in the NDT group compared to sham treatment (CTR OUT) group (F (2,26) = 5.28; *p* = 0.039; 95% CI: 0.01–0.39), and in the CTR IN group compared to CTR OUT group (F (2,26) = 5.28; *p* = 0.017; 95% CI: 0.04–0.44).

Finally, NDT treatment induced a significant anti-apoptotic behaviour compared to the sham treatment (CTR OUT) (F (2,23) = 4.78; *p* = 0.02; 95% CI: 2.14–20.51; η_p_^2^ = 0.294, medium effect; Figure 7D), as demonstrated by a lower Bax/Bcl-XL ratio in the NDT group.

## 4. Discussion

The repeated and selective nerve tensioning or flossing techniques, of which NDT consists, are non-pharmacological treatments able to modulate pain and, in particular, mechanical neuropathic pain in humans and animals [28,29,30,31,32,33,34,35,36]. NDT has been demonstrated to induce pro-regenerative effects by promoting the neuritis outgrowth and cell differentiation in sensory and motor neurons in vitro in a dose-dependent manner and preventing apoptosis only in sensory neurons [44]. These processes were also observed in our ex vivo experiments on DRG explants. In addition, multilevel changes in the central nervous system have been reported in the literature. In particular, it has been demonstrated that NDT induces a significant reduction in the expression of TRPV1 channels, nerve growth factor (NGF), and glial fibrillary acidic protein (GFAP) in the homologous DRG and spinal cord [58,78], and increases the expression of the µ-opioid receptor in the DRG [79]. In a rat neuropathic pain model, NDT affected the expression of GFAP and brain-derived neurotrophic factor (BDNF) proteins in ventral posterolateral thalamus nucleus and periaqueductal grey [58] and the k-opioid receptor expression in the periaqueductal grey [59]. Notably, all these processes are related to pain modulation, but none of them have been defined to be selective for mechanical allodynia and nerve regeneration. Our aims were therefore to assess the effects of NDT on sensory motor recovery and mechanical neuropathic pain modulation and its possible involvement in nerve regeneration after a nerve injury.

The current study shows that NDT stimulates nerve regeneration by restoring, significantly early, the nociception in the treated animals with a significantly higher performance of the upper limb in a high demanding task such as climbing a rope, without affecting fine movement dexterity or muscle strength of finger flexors. We have also defined that NDT promotes neuronal cell survival in the DRG cultures, speeds up neurites outgrowth and reduces intraneural fibrosis, according to the available evidence reported in the literature on NDT used in clinical and preclinical studies [36,42,48,58,59,78,79,80,81]. In addition, our data show that NDT prevents ipsilateral and contralateral mechanical threshold lowering induced by the nerve crush injury, which is a phenomenon sustained by a multilevel PNS plasticity and systemic processes [82,83,84,85].

Contrary to other studies on NDT described in the literature, in which animals under anaesthesia are subjected to NDT protocols [48,58,59,78,79,80], in our study, we deliberately kept the animal awake. This allows the clinician to feel the tissue resistance, including muscle contraction, as it happens in the clinical setting to properly tailor the treatment with real-time feedback. In addition, it was possible to determine the nerve mechanical irritability by recording the number of paw withdrawals to repeated high-frequency nerve tensions, which showed significantly increased irritability of the neural tissue to mechanical stress only in the not treated group. At the time this test was administered, sensory recovery for touch and pain was restored for all groups (Figure 2 and Figure 3), and the test frequency, five times higher than treatment, was able to elicit temporal summation [56,57]. These data confirm that NDT can promote pain descending modulation effects as confirmed by an increased expression of the µ-opioid receptor in the DRG and k-opioid receptor in the periaqueductal grey detected in neuropathic pain models of rats treated with NDT protocols [59,79].

Our data from ex vivo experiments show that the NDT induces an anti-apoptotic effect in the DRG neurons as demonstrated by a lower Bax/BclxL ratio compared to the sham treatment (CTR OUT). Taken together, these data confirm that the NDT is well tolerated without any risk of side or negative effects on nerve healing processes or animal discomfort.

Notably, our results show that NDT administered after nerve injury downregulates *TACAN* expression, but not *TLR2* (selectively related to immune-mediated mechanical allodynia) [15], in the ipsilateral and contralateral DRG. Moreover, our results on ex vivo DRG explants confirmed *TACAN* downregulation induced by NDT treatment. These data support the hypothesis that *TACAN* expression in the DRG gives a direct indication of nociceptive neurons activation after a nerve injury [10] and that NDT effectively modulates MA and hyperalgesia mechanisms in sensory neurons.

Since Sellheim pioneered the paravertebral block technique in 1905 to suppress local pain before and after surgery [86], the peripheral nerves and the DRG have become a therapeutic target for pain suppression [87,88,89]. Only the recent discovery of gene regulation in the DRG and the studies on intraforaminal lidocaine injection on short-term phantom limb pain suppression [10,15,90,91,92] led to the hypothesis that the DRG can be a pain driver also responsible for pain-gating effects. A growing body of evidence shows that DRG stimulation induces a reduction in pain responses to mechanical stimuli [90,92,93,94,95,96]. Even if the DRG was not included in the pain gate control theory postulated by Melzack and Wall [97], the short- and long-term effects reported in our ex vivo and in vivo experiments, in agreement with the literature, confirm that changes observed in the DRG were related to mechanical pain modulation in animals. For these reasons, we suggest that the DRG plays a relevant role in the pathophysiology of pain, and not only nociception, especially when the pain-related phenomenon is stimuli-dependent such as allodynia and hyperalgesia. In addition, we have shown that the administration of tailored tensions protocol to the nerves affects stimuli-dependent pain by affecting selectively the high threshold mechanosensitive receptor expression in the DRG.

In this study, we also pointed out the possible involvement of NDT in nerve regeneration processes. The morphometric analysis had shown that the number of myelinated fibres is similar between injured and not injured sides only in treated groups, even if they are significantly smaller as normally occurs in regenerating nerves. Notably, only the post-treated group had shown, after injury, a significantly higher maturation of the myelinated fibres (g-*ratio*) and a lower amount of intraneural collagen deposit than the NO NDT group. According to our data, Lima and colleagues [98] have shown that NDT significantly reduces the intraneural scar tissue formation after sciatic nerve crush injury. We suggest that a plausible NDT effect on intraneural collagen deposit possibly leads to lower intraneural pressure that causes aberrant or irregular low-rate nociceptor discharge, as shown by Bove and co-workers [99], and positively affects nerve regeneration.

The NDT positive effects were also confirmed by ex vivo experiments. One session of 30 repeated strains, administered for 3 min, induced a higher neuritis length of about one millimetre when compared to sham treatment (CTR OUT). These morphological changes together with our previous in vitro studies [44] and behavioural tests from animals suggest that NDT promotes a faster nerve regeneration after a nerve injury with tactile and nociception early recovery.

These pro-regenerative effects may be crucial for the injured nerve to prevent distal atrophic processes and to reach the target tissue to recover normal sensibility or muscular performance in the case of sensory or motor neurons as in the case of the present experiment [48,100,101,102]. The results of the motor tests revealed that NDT prevents gross motor function deficits observed at the rope test without compromising the dexterity and strength of the forelimb, at the grasping and grissini test, over time [55,60,61,62].

The NDT modulates pain in pre-clinical and clinical studies, showing effects only on the treated side [80,103]. Our data provide evidence that NDT prevents sensitization to normal mechanical stimuli that we observed by the significantly lowered mechanical threshold in the untreated rats on both sides, suggesting that NDT may activate metameric or systemic processes, avoiding nerve sensitization to mechanical stimuli. Even if some pain behaviours related to contralateral DRG changes after a nerve injury, as described in the literature [82,84,85,104], only a few clinical studies have shown in healthy subjects that NDT improves the mobility of the contralateral untreated limb [105,106]. We provide evidence that NDT administered before nerve injury modulates pain with a significant antiallodynic effect for mechanical stimuli 19 days after nerve injury, with no significant difference in *TACAN* gene expression in the ipsilateral and contralateral DRG assessed 24 days after the injury when compared to controls. Results obtained have also shown that NDT pretreatment induces a delayed sensory recovery for mechanical pain stimuli at 12 days instead of 8 days obtained in the NDT POST group. Similar beneficial effects between pre- and post-NDT treatment on motor tasks were detected, except for the speed in the grissini piece consumption which was significantly higher in the pretreated than in the control group. For these reasons, it is worth further exploring the mechanisms induced by NDT before the injury and studying those that are involved in sensory modulation at the contralateral side of nerve injury.

Despite the encouraging data obtained on nerve regeneration and mechanical pain modulation processes induced by the NDT also observed in other experiments on rat models of neuropathic pain [59,78,80], it is worth be reminded of the neuropathic pain complexity. This is a phenomenon involving multiple-level changes in the peripheral and central nervous systems affecting the immune and endocrine systems [56,57,83,84,85]. Our assessments were limited at exploring the mechanisms of the stimuli-dependent phenomena of MA and hyperalgesia, but neuropathic pain is also responsible for other disabling conditions such as persistent pain and nocturnal pain. The NDT protocol developed in our study has been shown to be well tolerated on awake animals with dosages and posology suitable for clinical application. However, future clinical studies are necessary to perform to establish its effectiveness compared to other protocols already adopted in the clinical settings for neuropathic pain patients.

## 5. Conclusions

Our study shed light on how NDT is an intervention able to promote nerve regeneration and mechanical pain suppression, which currently is still a common condition particularly difficult to treat with existing therapies. We have also shown that the NDT affects a specific pathophysiological mechanism responsible for mechanical allodynia and hyperalgesia by modulating the *TACAN* expression and not *TLR2*. We provided evidence that NDT is not a time-consuming intervention for neuropathic pain treatment, and we suggest starting the administration early after acute nerve injury (2–4 days after trauma). Our data, together with those available in the literature, showed that NDT is an effective non-pharmacological intervention for nerve injuries and neuropathic pain management. In addition, since it does not require any electronic device or expensive tool, it is also a sustainable intervention affordable for any health care system worldwide, with negligible environmental impact.

No data were available in the literature to define the proper dosage and posology of NDT treatment to be adopted in clinical settings, increasing the risk of under or over-treating the patient. We believe it is worth translating the NDT protocol described in our experiments in the clinical setting to treat the diseases in which NDT is already effective, such as neck and back pain, and future randomized controlled trials could define the effectiveness of this protocol compared to those already adopted in the literature. In addition, we suggest the potential role of NDT in the treatment of nerve injuries, which is a field in which nerve mobilization could have an impact on preventing chronicity and providing beneficial effects. Neuropathic pain conditions involving peripheral nerve damages in the limbs and/or trunk, such as radiculopathies or diabetes, have already shown improvements after NDT administration; it is worth exploring with future randomized clinical trials its effects on other types of acquired neuropathic pain conditions in which MA and hyperalgesia are involved.

## Figures and Tables

**Figure 1 biomedicines-10-01296-f001:**
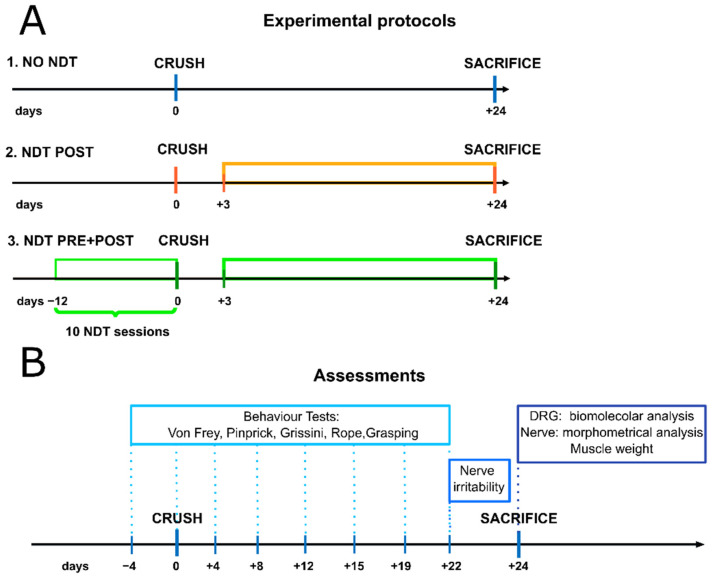
Treatment protocols and timeline of the assessments. (**A**) The coloured lines of the figure represent the duration of NDT protocols administrated 5 days/week, 30 sessions each day of treatment in orange (NDT POST) and green (NDT PRE + POST). (**B**) Assessments performed at each time point are reported in the coloured frames.

**Figure 2 biomedicines-10-01296-f002:**
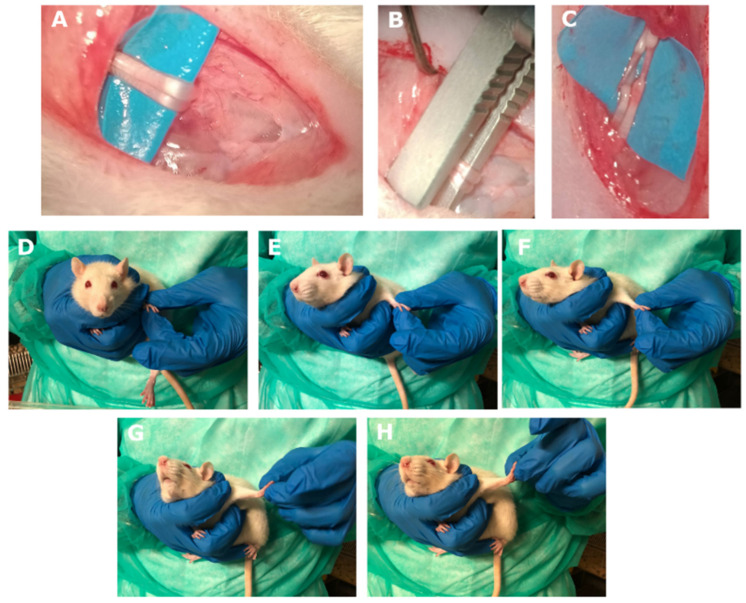
Crush injury protocol (**A**–**C**). (**A**) After a 2 cm skin incision along the medial aspect of the long axis of the arm above the elbow, the median and ulnar nerves were isolated and exposed. (**B**) Median and ulnar nerves were crushed with a non-serrated clamp for 30 s. (**C**) The crushed area was checked to make sure there was no loss of continuity of the connective components of the nerves. Neurodynamic tensioning treatment (**D**–**H**). NDT was administered to the injured side only (left side) to awake animals. The neurodynamic test from the neutral position (**D**) consisted of contralateral neck side flection (**E**), shoulder abduction and elbow extension (**F**), wrist extension (**G**), and fingers extension (**H**) performed until the clinician perceived resistance to mobilization.

**Figure 3 biomedicines-10-01296-f003:**
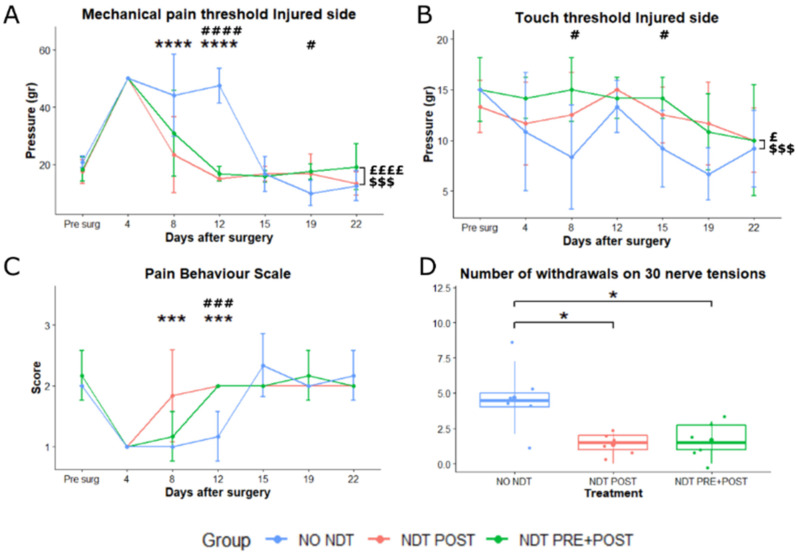
Sensory assays. (**A**,**B**) Mechanical pain threshold and touch threshold at the paw of the median and ulnar nerve crush injured side obtained by Von Frey monofilament administration at each experimental time point (x-axis) and expressed in grams (g). (**C**) Pain behaviours at the Pinprick test were recorded when four consecutive pin administrations are given to the forepaw. (**D**) The number of withdrawals, at 22 days after injury, was recorded during 30 consecutive neurodynamic tests of the upper limb. Data are expressed as mean ± SD. The analysis compared all groups to the control group (NO NDT) with ANOVA for repeated measures (data are normally distributed with comparable variances). Differences at each time point between groups are reported * *p* < 0.05, *** *p* < 0.0001, and **** *p* < 0.0000, comparing the group NDT POST to the NO NDT group, or #, ### and ####, *p* < 0.05, *p* < 0.001 and *p* < 0.000, comparing the NDT PRE + POST to the NO NDT group. Overall differences between groups are reported £ *p* < 0.05 ££££ *p* < 0.000 comparing the NDT POST to the NO NDT group and $$$ *p* < 0.001 comparing the NDT PRE + POST to the NO NDT group. n = 18; 6 for each experimental group.

**Figure 4 biomedicines-10-01296-f004:**
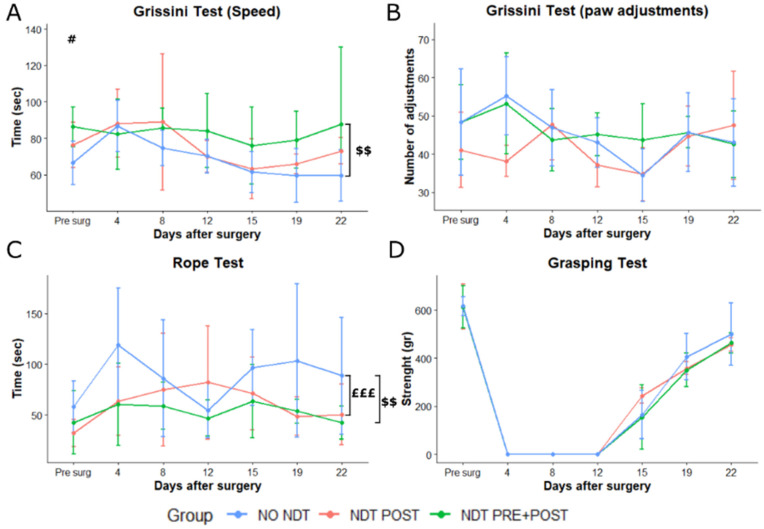
Motor assays. (**A**,**B**) Speed and number of paw adjustments recorded while rats were eating 7 cm grissini pieces, at different time points after nerve injury and repair. (**C**) Speed in climbing a 160 cm vertical rope expressed in seconds (sec). (**D**) Grasping strength was recorded by the device as maximal resistance to pull the grid of the device with the injured side paw. Data are expressed as mean ± SD. The analysis compared all groups to the control group (NO NDT) with ANOVA for repeated measures (data are normally distributed with comparable variances). Differences at each time point between groups are reported # *p* < 0.05, comparing the NDT PRE + POST to the NO NDT group. Overall differences between groups are reported £££ *p* < 0.001 comparing the NDT POST to the NO NDT group and $$ *p* < 0.01 comparing the NDT PRE + POST to the NO NDT group. Detailed data of the statistical analysis are reported in Appendix A. n = 18; 6 for each experimental group.

**Figure 5 biomedicines-10-01296-f005:**
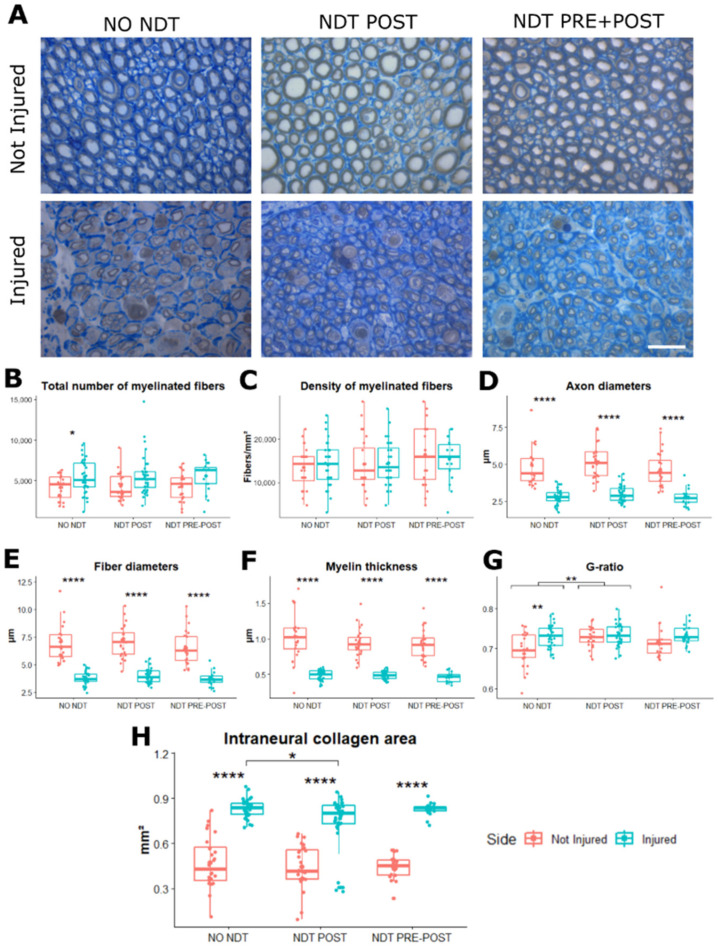
Morphological and morphometric analysis. (**A**) High magnification light microscopy representative images of toluidine blue-stained semi-thin cross-sections of uninjured median nerves and injured median nerves after 24 days from the lesion, for each treatment. Scale bar: 20 µm. (**B**–**H**) Stereological assessment of morphological changes during nerve regeneration. (**B**) Number of myelinated fibres; (**C**) myelinated fibre density; (**D**) axon diameters; (**E**); fibre diameters; (**F**) myelin thickness; (**G**) axon diameter/fibre diameter *ratio* (g-*ratio*); (**H**) intraneural collagen area. Data are expressed as mean ± SD. The analysis compared all groups and sides to the control group (NO NDT) with a Two-way ANOVA (data are normally distributed with comparable variances). Differences are reported * *p* < 0.05, ** *p* < 0.01, and **** *p* < 0.0000. n = 18; 6 for each experimental group.

**Figure 6 biomedicines-10-01296-f006:**
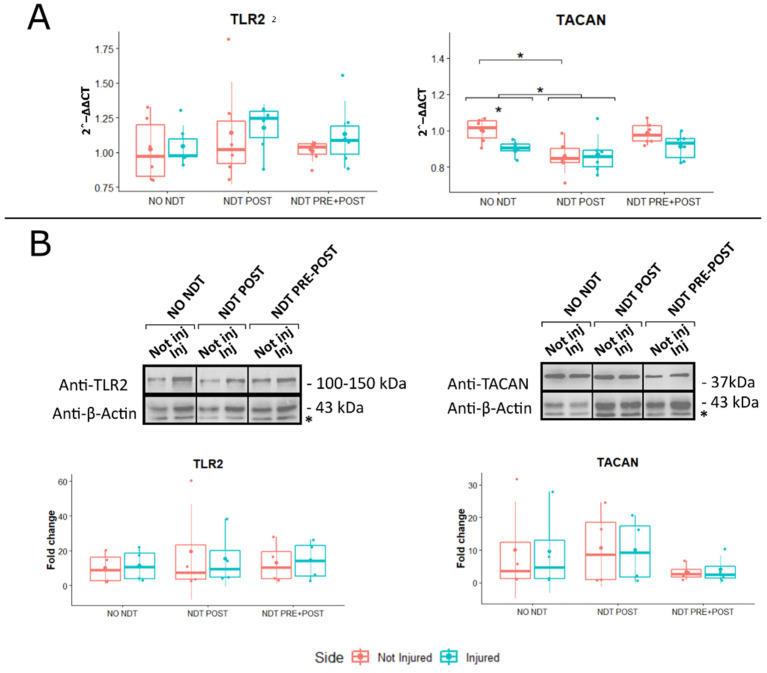
Gene and protein expression analysis of markers linked to mechanical allodynia and apoptosis expressed in the DRG. Panel (**A**): Beneficial or side effects induced by the neurodynamic protocols on DRG were assessed adopting the relative quantification (2-ΔΔCt) of genes by qRT-PCR. Data normalization was performed considering TATA-box binding protein (TBP) as a housekeeping gene. All data were calibrated to NO NDT Not Injured samples. Panel (**B**): A representative Western blot is shown; actin was used as a loading control. Asterisks (*) in panel B identify unspecific bands. Values in the graphics are expressed as mean ± SD. Respectively, two-way and one-way ANOVA were carried out (data are normally distributed with comparable variances). * *p* < 0.05. n = 18; 6 for each experimental group.

**Figure 7 biomedicines-10-01296-f007:**
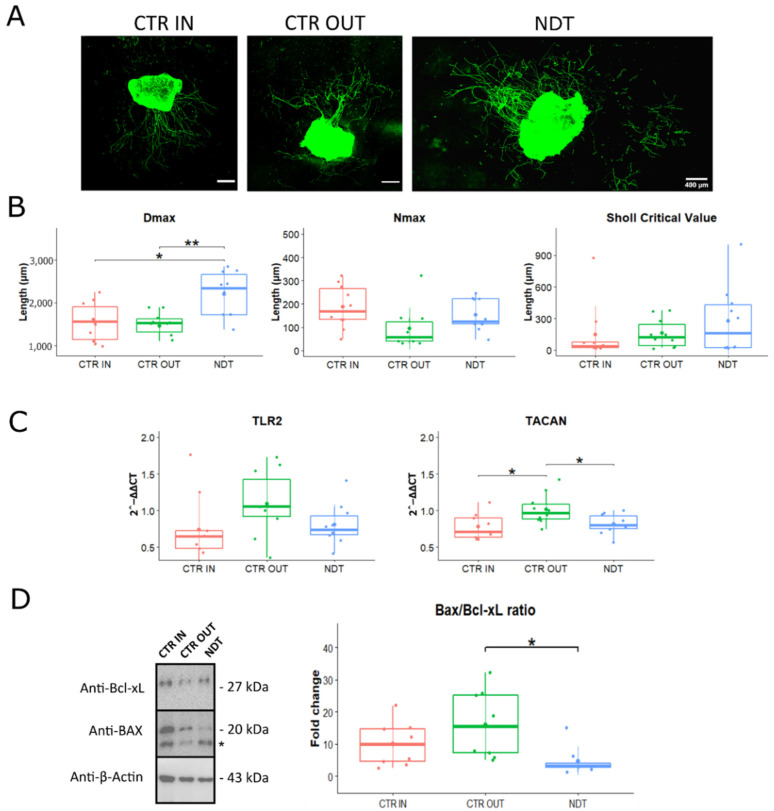
Effects of neurodynamic treatment on cell morphology, gene and protein expression on dorsal root ganglia (DRG) explants. Panel (**A**): Representative images of rat organotypic dorsal root ganglia neurons stained with βIII-tubulin reported for each type of experimental protocol. Scale bar: 400 µm. Panel (**B**): Quantitative analysis of the distance of the longest neurite (Dmax), the maximum number of neurites (Nmax), and the Sholl critical value, defined as the distance from the organotypic culture centre. Values in the graphics are expressed as mean ± SD. Panel (**C**): Gene expression analysis of markers linked to mechanical allodynia and neuropathic pain. Beneficial or side effects induced by the neurodynamic protocols on dorsal root ganglia explants were assessed adopting the relative quantification analysis (2^−ΔΔCt^) of genes by qRT-PCR. Data normalization was performed considering TATA-box binding protein (TBP) as a housekeeping gene. All data were calibrated to CTR OUT sample). Panel (**D**): BAX and Bcl-xL protein expression in DRG explants. Protocols are described as follows: not treated (CTR IN), sham-treated (CTR OUT), and treated (NDT) with 30 repetitions of neurodynamic treatment. Experiments were carried out in a biological octuplicate (n = 21 for a technical replicate). Asterisks (*) in panel (**D**) identify unspecific bands. Values in the graphics are expressed as mean ± SD. For normally distributed data with comparable variances One-way ANOVA was carried out, while Kruskal–Wallis test was used for nonparametric data; asterisks show statistically significant differences with CTR OUT (sham sample); * *p* < 0.05, and ** *p* < 0.01.

**Table 1 biomedicines-10-01296-t001:** Sequences of primers used for quantitative real-time PCR.

*Gene*	*Sequence*	*Amplicon Length (bp)*	*Accession Number*
** *TLR2* **	forward: 5′-CAAACTGGAGACTCTGGAAGCAGG-3′ reverse: 5′-CACACAGGTAGCTGTCTGCC-3′	125	NM_198769.2 NM_011905.3
** *TACAN* **	forward: 5′-TGCAGCAGGACTTCCAAGGTATCC-3′ reverse: 5′-CGCTTCTTCTGGCGTGTGATAGAG-3′	115	NM_001010945.1

**Table 2 biomedicines-10-01296-t002:** Antibodies used for Western blot analysis.

** *Primary Antibodies* **				
	**Code**	**Dilution**	**Host**	**Source**
**Actin**	A5316	1:4000	Mouse	Sigma-Aldrich
**Bax**	SC-23959	1:600	Rabbit	Santa Cruz Biotechnology Inc.
**Bcl-xL**	SC-8392	1:600	Mouse	Santa Cruz Biotechnology Inc.
**TACAN**	17455–1-ap	1:1000	Rabbit	Proteintech Europe
**TLR-2**	SC-166900	1:300	Mouse	Santa Cruz Biotechnology Inc.
** *Secondary Antibodies* **				
	**Code**	**Dilution**	**Host**	**Source**
**HRP Conjugated-anti-rabbit**	7074	1:15,000	Goat	Cell Signalling
**HRP Conjugated-anti-mouse**	7076	1:15,000	Goat	Cell Signalling

## Data Availability

The datasets used and/or analysed during the current study are available from the corresponding author on reasonable request.

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
