# Peer review of "Neurodynamic Treatment Promotes Mechanical Pain Modulation in Sensory Neurons and Nerve Regeneration in Rats"

_biomedicines, 2022, doi:10.3390/biomedicines10061296_

Round 1

Reviewer 1 Report

The authors objective in this research has been to define in vivo and ex vivo characteristics on how the neurodynamic treatment (NDT) could promote nerve regeneration and modulate some processes in the DRG linked to MA and hyperalgesia. The experiments have been conducted in Wistar rats after unilateral median and ulnar nerve crush. The therapeutic effects of NDT was investigated. The authors also investigated the possible protective effects of NDT which was administered 10 days 25 before the injury.

The authors have employed several techniques in this research work including behavioral tests, morphological and morphometrically analysis, gene and protein expression analysis. The ex vivo experiments demonstrated that NDT increases neurite regrowth and confirmed the modulation of TACAN, which is a mechanical high threshold ion channel that can be activated only by a cell surface tension higher than 35 mN/m and is responsible for mechanical painful stimuli transduction. This research showed that the NDT can have potential to be adopted in clinical practice after nerve injuries to promote functional recovery and pain modulation.

The authors are encouraged to clarify these points in the revised version:

Please add the reason for using only female animals. ARRIVE guidelines encourages having both sexes included.

Please add the limitation of this study and challenges for translating these data and findings to clinical use in humans.

Please moderate the conclusion as there is a "potential" for NDT in clinic. We need evidence first that it works and it is safe. 

Author Response

We thank the reviewer for carefully reviewing our manuscript and for her/his constructive criticisms. We are grateful for the impulses given to us and have carefully revised the manuscript in accordance to the reviewer’s comments. Today we present an improved, more clear, revised version of our manuscript and provide below a point-to-point response to the reviewer’s comments.

Reviewer’s question: Please add the reason for using only female animals. ARRIVE guidelines encourages having both sexes included.

Answer: We agree with the reviewer. Although the guidelines encourage the inclusion of both sexes in animal experiments, we decided to conduct this research using female rats as experimental model, primarily because they maintain, in adulthood, a limited weight and size and this has allowed an easier handling during NDT administration and functional testing. Furthermore, RNA-Seq studies on DRG after nerve injuries in rats have shown that the genes related to mechanical allodynia that we have assessed in our study are similarly expressed both in males and females [25–27].

-Changes made to the MS:

Paragraph 2.2.1. “In vivo surgical and treatment protocols” of the materials and methods section has been modified, adding details about the choice of the sex of the animal models (Line 96, highlighted in yellow).

Only female animals were adopted since they maintain, in adulthood, a limited weight and size and this has allowed an easier handling during NDT administration and functional testing. Furthermore, RNA-Seq studies on DRG after nerve injuries in rats have shown that the genes related to mechanical allodynia that we have assessed in our study are similarly expressed both in males and females [25–27]

Reviewer’s question: Please add the limitation of this study and the challenges for translating these data and findings to clinical use in humans.

Answer: We modified the text according to the reviewer’s comment. We inserted a paragraph as detailed below to highlight the limitation of the study and the clinical future perspectives.

-Changes made to the MS:

Paragraph 4. Discussion section.  We have added a paragraph highlighting the limitations of the study (Line 681, highlighted in yellow).

   “Despite the very encouraging data obtained on nerve regeneration and mechanical pain modulation processes induced by the NDT also observed in other experiments on rat models of neuropathic pain [59,78,80] it is worth reminding the neuropathic pain complexity. Indeed, this is a phenomenon involving multiple-level changes in the peripheral and central nervous systems, affecting the immune and endocrine systems [56,57,83–85].Our assessments were limited at exploring the mechanisms of the stimuli-dependent phenomena of MA and hyperalgesia, but neuropathic pain is also responsible for other disabling conditions like persistent pain and nocturnal pain. The NDT protocol developed in our study had shown to be well tolerated on awake animals with dosages and posology suitable for clinical application. However, future clinical studies are necessary to be performed to establish its effectiveness compared to other protocols already adopted in the clinical settings for neuropathic pain patients.“

Reviewer’s question: Please moderate the conclusion as there is a "potential" for NDT in clinic. We need evidence first that it works and it is safe. 

Answer: We thank the reviewer for the comment, we tried to moderate our conclusions by highlighting the need of demonstrating the effectiveness of NDT in clinic.

-Changes made to the MS:

Abstract: Conclusion. (Line 33, highlighted in yellow)

The results obtained in this study on the biological and molecular mechanisms induced by NDT will allow the exploration, in future clinical trials, of its efficacy in different conditions of neuropathic pain

Conclusion section. Paragraph 5. (Line 708, highlighted in yellow)

“We believe it is worth translating the NDT protocol described in our experiments in the clinical setting to treat the diseases in which NDT is already effective like neck and back pain, and future randomized controlled trials could define the effectiveness of this protocol compared to those already adopted in the literature. Also, we suggest the potential role of NDT in the treatment of nerve injuries which is a field in which nerve mobilization could have an impact on preventing chronicity and providing beneficial effects. Neuropathic pain conditions involving peripheral nerve damages in the limbs and/or trunk, like radiculopathies or diabetes, have already shown improvements after NDT administration; it is worth exploring with future randomized clinical trials its effects also on other types of acquired neuropathic pain conditions in which MA and hyperalgesia are involved.”

Reviewer 2 Report

REVIEWER’S COMMENTS

The manuscript The neurodynamic treatment promotes mechanical pain modulation in sensory neurons and nerve regeneration in rats.by Carta et al suggests that the NDT can be safely adopted in clinical practice after nerve injuries to promote functional recovery and pain modulation.

 1.     Abstract: lines 17-20  and line 28 are grammatically incorrect and need to be re-written so that the information is clear.

a.  Lines 17-20 - please re-write as two sentences. Please put a period after hyperalgesia.

b.     Line 28 - Please re-write as “Behavioral tests, morphological and morphometrical analysis, gene and protein expression analysis were performed, and these tests revealed that NDT……”

2.     Line 91: The official SI unit symbol is “g” not “gr”. Please fix this wherever applicable in this paper.

3.     Please include the catalog numbers of all the reagents used in this study.

4.     Please include “n” in the figure legends.

5.  Please improve the resolution of the figures. Resolution should be at least 300 dpi.

6.     Please briefly discuss the potential future directions that could be a follow-up to this study.

Author Response

We thank the reviewer for carefully reviewing our manuscript and for her/his constructive criticisms. We are grateful for the impulses given to us and have carefully revised the manuscript in accordance to the reviewer’s comments. Today we present an improved, more clear, revised version of our manuscript and provide below a point-to-point response to the reviewer comments.

Reviewer question: Abstract: lines 17-20 and line 28 are grammatically incorrect and need to be re-written so that the information is clear.

  1. Lines 17-20 - please re-write as two sentences. Please put a period after hyperalgesia.
  2. Line 28 - Please re-write as “Behavioral tests, morphological and morphometrical analysis, gene and protein expression analysis were performed, and these tests revealed that NDT……”

Answer: We apologize for not being clear, we have rephrased the reported sentences.

-Changes made to the MS:

Lines 17-20: the sentence was changed as suggested by the reviewer “Somatic nerve injuries are a rising problem leading to disability associated with neuropathic pain commonly reported as mechanical allodynia (MA) and hyperalgesia. These symptoms are strongly dependent on specific processes in the dorsal root ganglia (DRG).

Line 28: the sentence was changed as suggested by the reviewer “Behavioral tests, morphological and morphometrical analysis, gene and protein expression analysis were performed, and these tests revealed that NDT……”

Reviewer’s question: Line 91: The official SI unit symbol is “g” not “gr”. Please fix this wherever applicable in this paper.

Answer: We apologize for the error. We have replaced the symbol where necessary in the manuscript (highlighted in yellow).

Reviewer’s question: Please include the catalog numbers of all the reagents used in this study.

Answer: We have included the catalog numbers of all the reagents in the materials and methods section manuscript (highlighted in yellow)

Reviewer’s question: Please include “n” in the figure legends.

Answer: We have included in the figure legends (highlighted in yellow)

Reviewer’s question: Please improve the resolution of the figures. Resolution should be at least 300 dpi.

Answer: We have increased the resolution of the images.

Reviewer’s question: Please briefly discuss the potential future directions that could be a follow-up to this study.

Answer: We modified the text according to the reviewer’s comment. We inserted a paragraph as detailed below to highlight the limitation and the clinical future perspectives.

-Changes made to the MS:

Paragraph 4. Discussion section.  We have added a paragraph highlighting the limitations and clinical future perspectives of the study (Line 681, highlighted in yellow).

   “Despite the very encouraging data obtained on nerve regeneration and mechanical pain modulation processes induced by the NDT also observed in other experiments on rat models of neuropathic pain [59,78,80] it is worth reminding the neuropathic pain complexity. Indeed, this is a phenomenon involving multiple-level changes in the peripheral and central nervous systems, affecting the immune and endocrine systems [56,57,83–85]. Our assessments were limited at exploring the mechanisms of the stimuli-dependent phenomena of MA and hyperalgesia, but neuropathic pain is also responsible for other disabling conditions like persistent pain and nocturnal pain. The NDT protocol developed in our study had shown to be well tolerated on awake animals with dosages and posology suitable for clinical application. However, future clinical studies are necessary to be performed to establish its effectiveness compared to other protocols already adopted in the clinical settings for neuropathic pain patients. “